# UNDERSTANDING PARAMETER SALIENCY VIA EXTREME VALUE THEORY

## ABSTRACT

Deep neural networks are being increasingly implemented throughout society in recent years. It is useful to identify which parameters trigger misclassification in diagnosing undesirable model behaviors. The concept of parameter saliency is proposed and used to diagnose convolutional neural networks (CNNs) by ranking convolution filters that may have caused misclassification on the basis of parameter saliency. It is also shown that fine-tuning the top ranking salient filters efficiently corrects misidentification on ImageNet. However, there is still a knowledge gap in terms of understanding why parameter saliency ranking can find the filters inducing misidentification. In this work, we attempt to bridge the gap by analyzing parameter saliency ranking from a statistical viewpoint, namely, extreme value theory. We first show that the existing work implicitly assumes that the gradient norm computed for each filter follows a normal distribution. Then, we clarify the relationship between parameter saliency and the score based on the peaks-over-threshold (POT) method, which is often used to model extreme values. Finally, we reformulate parameter saliency in terms of the POT method, where this reformulation is regarded as statistical anomaly detection and does not require the implicit assumptions of the existing parameter-saliency formulation. Our experimental results demonstrate that our reformulation can detect malicious filters as well. Furthermore, we show that the existing parameter saliency method exhibits a bias against the depth of layers in deep neural networks. In particular, this bias has the potential to inhibit the discovery of filters that cause misidentification in situations where domain shift occurs. In contrast, parameter saliency based on POT shows less of this bias.

## 1 INTRODUCTION

Deep learning models can perform a variety of tasks in computer vision with high accuracy. Despite their adoption in many applications, we usually do not have an understanding of the model's decision making process. This means there is a potential risk when we use deep learning models for high-stakes applications. Conventional research on the explainability of deep learning models in computer vision has focused on generating a saliency map that highlights image pixels inducing a strong response from the model(Selvaraju et al., 2017; Simonyan et al., 2013; Sundararajan et al., 2017; Fong & Vedaldi, 2017; Petsiuk et al., 2018). Although this kind of visualization often makes intuitive sense for humans and partly explains the model behavior, a saliency map is helpless for fixing an incorrect classification result because it is not linked with the parameter space. Recently, Levin et al. (2021) proposed ranking convolutional filters according to a score called *parameter saliency* for exploring the cause of CNN misclassifications. The parameter saliency reflects strong filter importance determined by the normalized gradient, and the top-ranked filters are shown to have a greater relationship with the classification result when modifying the filters. However, there is a knowledge gap as to why parameter saliency ranking can find filters inducing misidentification. Additionally, we found in our preliminary experiments that the ranking algorithm has a bias against the depth of layers in deep neural networks, which can lead to the model yielding mediocre outcomes in certain situations.

To address the bias problem, we elucidate the concept of parameter saliency from a different perspective. We first formulate the problem of ranking salient filters in terms of statistical anomaly detection for parameter-wise saliency profiles. We then analyze the relationship between salient filter ranking

and the peaks over threshold (POT) (Pickands III, 1975; Grimshaw, 1993) method based on extreme value theory (EVT) (Haan & Ferreira, 2006) and show that the existing method can be viewed as a special case of our formulation based on EVT under appropriate assumptions on the gradient distribution. EVT, a branch of statistics that emerged to handle the maximum and minimum values of a sequence of data, enables us to estimate the probability of extreme events observed in the tail of probability distributions.

For the experiments in this work, we compared the effects of modifying salient filters detected by the existing method and the POT-based method using the same metrics as the original work(Levin et al., 2021). To further investigate the properties of our reformulaiton, we used datasets such as MNIST and SVHN in which domain shift occurs and analyzed the top-ranked filter distribution to clarify the relationship between salient filters and insufficient feature extraction.

In summary, we have made the following contributions.

- We reformulate salient filter ranking as statistical anomaly detection in which parameter saliency is interpretable as the probability of observing an event.
- We clarify the relationship between salient filter ranking and the POT method in EVT.
- We demonstrate that the POT method operates well even when domain shift occurs, while an intrinsic bias in the baseline method prevents consistent performance.

## 2 RELATED WORK

**Interpretability and Explainability of Machine Learning Models**   There are two main approaches to understanding machine learning models: using intrinsically interpretable models or using post hoc methods(Molnar, 2022). Models of the first type have a restricted form of architectures, e.g., decision trees(Frosst & Hinton, 2017) and linear models, that make it possible to interpret the calculation process. In contrast, the second type of methods are open to arbitrary models and explain why the model behaves in a specific manner. Counterfactual explanation(Verma et al., 2020) and LIME(Ribeiro et al., 2016) are two representative examples of this type.

**Deepening the Understanding of CNNs**   CNNs(Simonyan & Zisserman, 2014; He et al., 2016) have shown an outstanding performance in various computer vision tasks, but they are innately black boxes. To alleviate this problem, many saliency map generation methods have been proposed to visualize which image pixels are sensitive to the models. Some methods make maximum use of gradient information(Selvaraju et al., 2017; Simonyan et al., 2013; Sundararajan et al., 2017), while others perturb or blur the original image to quantify the effect of pixels on classification(Fong & Vedaldi, 2017; Petsiuk et al., 2018). Various criteria have been proposed to evaluate the quality and guarantee the validity of saliency maps including sanity check(Adebayo et al., 2018), relevance to the output score(Samek et al., 2016), and user experience(Alqaraawi et al., 2020). Another line of work focuses on the roles of convolutional layers and shows that CNNs work as a feature extractor(Bau et al., 2017; Zeiler & Fergus, 2014).

**Importance in parameter-space**   Pruning for CNN model compression is closely related to the importance of convolutional filters. Filter importance is estimated via the activation response(He et al., 2022), the $l_1$ norm of the filter weights (Li et al., 2016), group lasso regularization(Wen et al., 2016), neuron importance score propagation(Yu et al., 2018), and the mean gradient criterion(Liu & Wu, 2019). Alternative directions using the importance include updating only a subset of parameters with top-N importance(Sun et al., 2017) and retraining a model by referencing possibly better parameters(Zhang & Chan, 2019). This kind of work is not limited to computer vision. For example, in natural language processing, the linguistic roles of neurons have been explored(Bau et al., 2018).

## 3 PRELIMINARY

### 3.1 PARAMETER SALIENCY

In this section, we briefly review parameter saliency proposed by Levin et al. (2021). Let $(x, y) \in (\mathcal{X}, \mathcal{Y})$ be a pair of a sample and its ground-truth label in a dataset, where $\mathcal{X}$ is the input space and

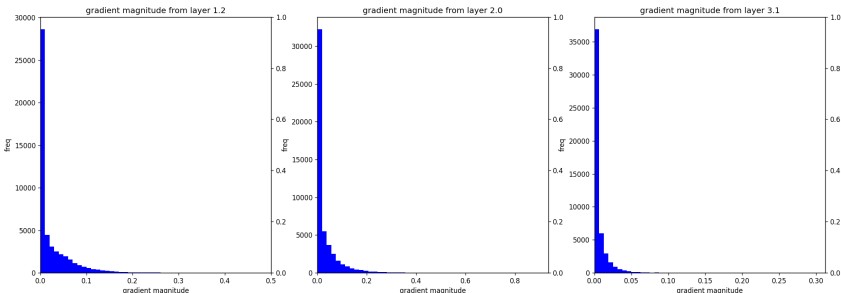

Figure 1: Distributions of gradient magnitude from different layers in ResNet-50.

$\mathcal{Y}$ is the corresponding set of classes. A model with parameters $\theta$ can be defined as a function $f_\theta : \mathcal{X} \rightarrow \mathcal{Y}$. In most cases, a model is trained so that $f_\theta$ minimizes a loss function $\mathcal{L} : \mathcal{F} \times \mathcal{X} \times \mathcal{Y} \rightarrow \mathbb{R}$, where $\mathcal{F}$ is the set of models $f : \mathcal{X} \rightarrow \mathcal{Y}$. Our goal is to identify which subset of $\theta$ caused the model's misclassification. Although there are various model architectures for different tasks, we mainly discuss how things work on CNNs.

On the hypothesis that parameters with a large gradient magnitude are important, the parameter-wise saliency profile is defined by $s_{\theta_i}(x, y) := |\nabla_{\theta_i} \mathcal{L}(f_\theta, x, y)|$, where $\theta_i \in \mathbb{R}$ is the $i$-th element of the parameter. Each convolutional filter is a subset of the parameters involved in feature extraction, so averaging the parameter-wise saliency profile within each convolutional filter gives us the filter-wise saliency profile:

$$\bar{s}(x, y)_j := \frac{1}{|\mathcal{I}_j|} \sum_{i \in \mathcal{I}_j} s_{\theta_i}(x, y), \tag{1}$$

where $\mathcal{I}_j$ is the index set of the parameters in the $j$-th convolutional filter.

Finally, we obtain the parameter saliency, or filter saliency in the case of CNN, by performing the z-score normalization. This normalization aims to find data-specific salient filters and avoid finding universally important filters. More precisely, we obtain filter saliency computed with $\mu_j$ and $\sigma_j$ which are the mean parameter-wise saliency profile and the standard deviation for the $j$-th filter over the validation set of a dataset such as ImageNet:

$$\hat{s}(x, y)_j := \frac{\bar{s}(x, y)_j - \mu_j}{\sigma_j}. \tag{2}$$

A higher value is considered to make a greater contribution to the misclassification and the ranking is formed by ordering filters so that filter saliency is in decreasing order. We can attribute misclassification results to parameters, and finding these parameters gives us a chance to diagnose a model and correct the model behavior.

### 3.2 INTRODUCTION OF THEOREM OF PICKANDS-BALKEMA-DE HANN

In this section, we explain the essential concept underlying EVT. We included a tutorial in Appendix C to supplement the minimum necessary knowledge of EVT.

EVT focuses on extreme values and the behavior of tail event and is useful for assessing the probability of rare events. It is often used to evaluate risks such as once-in-a-century flood risks or the probability of extreme losses in financial markets. The basic idea behind EVT is that the distribution of the largest observations from a large dataset converges to one of several specific types of extreme value distributions. Since using only the maximum value and ignoring the rest of the data result in the loss of information, the POT method was proposed as a common approach to investigating the relationship between the frequency and magnitude of extreme events where data points exceeding a certain threshold are considered to estimate the distribution of extremes.

The Pickands-Balkema-de Haan theorem(Pickands III, 1975; Balkema & De Haan, 1974) is the most relevant theorem to this paper.

**Theorem 1** (Pickands-Balkema-de Haan). *For a large class of random variables $X$, there exists a function $\beta(t) : \mathbb{R} \to \mathbb{R}$ such that*

$$\lim_{t \to \tau} \sup_{0 \le x < \tau - t} |\mathbb{P}(X - t \le x | X > t) - G(x | \alpha, \beta(t))| = 0, \tag{3}$$

*where $\tau \in \mathbb{R}$ is the finite or infinite right endpoint, and $G(x | \alpha, \beta(t))$ is the generalized Pareto distribution (GPD).*

Given a scale parameter $\alpha \in \mathbb{R}$ and a shape parameter $\beta \in \mathbb{R}$, a GPD is defined as follows:

$$G(x | \alpha, \beta) = \mathbb{P}(X \le x) = \begin{cases} 1 - \left(1 + \frac{\beta x}{\alpha}\right)^{-\frac{1}{\beta}} & \beta \ne 0, \\ 1 - \exp(-\frac{x}{\alpha}) & \beta = 0. \end{cases} \tag{4}$$

This theorem is called the second theorem in EVT and lays the foundation of the POT method. The method fits a GPD to the tail of the probability distribution with a sufficiently large threshold $T$, estimating $\mathbb{P}(X - T \le x | X > T)$. More specifically, suppose we have $N$ observations $X_1, X_2, \ldots, X_N$, where $X_i \in \mathbb{R}$, and $n$ out of $N$ observations exceed the threshold $T$. We denote their indices by $J_T$ and let $Y$ be the set of excesses over $T$. Mathematically, we have $J_T = \{i \,|\, X_i > T\}$ and $Y = \{X_i - T \,|\, i \in J_T\}$. We use maximum likelihood estimation with this $Y$ for finding the GPD. We also approximate $\mathbb{P}(X > T)$ with the empirical distribution function, i.e., $\mathbb{P}(X > T) \approx n/N$. As a result, we can estimate the probability of an observed value that is larger than the threshold $T$:

$$\mathbb{P}(X - T > x) = \mathbb{P}(X > T)\mathbb{P}(X - T > x | X > T) \approx \frac{n}{N}\left\{1 - G(x | \alpha, \beta)\right\}. \tag{5}$$

## 4   A CLOSER LOOK AT PARAMETER SALIENCY THROUGH THE LENS OF EVT

First, we describe the motivation for statistically interpreting the existing method. Next, we explain the reformulation of parameter saliency ranking as statistical anomaly detection. Finally, we provide a general formulation of parameter saliency ranking based on EVT.

### 4.1   MOTIVATION

In this work, we explore the following three questions.

1. **Does the distribution of each filter's saliency follow a normal distrbution?** It assumes in the z-score normalization that the data follow a normal distribution. However, the gradient norm may not be assumed to be normally distributed.

2. **Can each filter's saliency be used as a ranking score in the same line when each filter may follow a different distribution?** The normalized values from different distributions as in fig. 1 are used for sorting filters. However, different distributions have different probabilities of obtaining the same value; thus, the rankings can not necessarily reflect the authentic relation of anomalies among the filters.

3. **What bias would occur in the above case?** If the distribution is heavy-tailed, large data points occur relatively frequently. This significantly affects the sample mean and variance, which can be extremely large for certain samples due to these outliers and induce bias.

In seeking answers to the questions, we explain below how parameter saliency ranking can be understood in terms of statistical anomaly detection and EVT.

### 4.2   STATISTICAL INTERPRETATION

We provide a novel interpretation of parameter saliency ranking in terms of statistical abnormal detection where our goal is to identify the filters that have statistically more abnormal filter-wise saliency profiles. We first consider the statistical meaning of parameter saliency because it is reasonable to assume that an unusual saliency profile is formulated by the rarity of the value, i.e., the probability of taking the value of the saliency profile. More formally, we assume that filter-wise

saliency profile for the $j$-th filter $\bar{S}_j$ follows a probability distribution $\bar{S}_j \sim P_j(\bar{S}_j)$. Given an input that the model classified incorrectly, we compute the saliency profile for the $j$-th filter and obtain $\bar{s}_j$. Then, we construct a ranking of filters so that filters with a smaller value of $\mathbb{P}(\bar{S}_j > \bar{s}_j)$ are higher in the ranking.

We show below that comparing filter saliency is equivalent to comparing the probability of the observed filter-wise saliency profile under the assumption that the filter-wise saliency profile follows a normal distribution.

**Proposition 1.** *Suppose $\bar{S}_j$ is a random variable that follows the normal distribution $\mathcal{N}(\mu_j, \sigma_j)$ for any $j$, where $\mu_j \in \mathbb{R}$ and $\sigma_j \in \mathbb{R}$ equal to the mean and the standard deviation respectively. We define $\hat{S}_j$ by $\hat{S}_j = (\bar{S}_j - \mu_j)/\sigma_j$. Let $\bar{s}_j$ be a sample from each distribution and $\hat{s}_j$ be the normalized value of $\bar{s}_j$, i.e., $\hat{s}_j = (\bar{s}_j - \mu_j)/\sigma_j$. Then, for any pair of the normalized values $(\hat{s}_j, \hat{s}_{j'})$, the following holds:*

$$\hat{s}_j \leq \hat{s}_{j'} \iff \mathbb{P}(\bar{S}_j > \bar{s}_j) \leq \mathbb{P}(\bar{S}_{j'} > \bar{s}_{j'}). \tag{6}$$

The proof is in Appendix B. Proposition. 1 tells us that the baseline method compares the probability of a filter-wise saliency profile and becomes one solution in our formulation. However, the assumption required here might be too strong and unrealistic in practice, so we want to weaken the assumption.

## 4.3 PARAMETER SALIENCY ESTIMATION VIA POT

In revisiting our primary objective, we aim to identify the filters in a CNN that induce misclassification. To achieve this, we need a method that can quantitatively evaluate the filters inducing misclassification. Ideally, the metrics should **(i) be comparable across different layers using the same criteria**, **(ii) have few assumptions behind them**, and **(iii) be easily interpreted**. Here we seek an evaluation method that embodies these three ideal properties.

We assume that filters inducing misclassification for a particular image have unique characteristics specific to that image. These characteristics can be formulated as a higher probability of being an anomalous filter compared to other correctly classified images. This probabilistic representation seems rational for expressing abnormality and useful in terms of interpretability. Furthermore, when formalizing abnormality in terms of probability, it is common in statistical anomaly detection to use a tail probability, i.e., the probability that exceeds a specific threshold. In this case, EVT is more useful than traditional statistical methods.

EVT is designed to derive detailed insights about extreme values and their stochastic behavior from data with a limited sample size, in contrast to traditional statistical methods that require large samples to capture such features. Since extreme events, by their nature, are rarely observed, amassing a large amount of these events for analysis can be difficult. Similarly, the anomalous behavior of filters causing misclassification can also be considered a rare event. Furthermore, the POT method focuses on data points that surpass a specific threshold within a dataset. This enables us to estimate the probability of extreme events without using all the data points, thus maximizing the information extracted from a restricted sample.

In this work, we reformulate the rarity of each filter's saliency profile according to the probability $\mathbb{P}(X > x)$ by using the POT method, which we call *POT-saliency*.

Since EVT provide results for the tail behavior of various probability distributions, we can evaluate extreme value probabilities without assuming a specific distribution. Therefore, the most important advantage of this method is that it does not require the assumption of a normal distribution for the distribution when calculating a score for each filter, which allows for a unified analysis even among different strata using the same criteria.

**Require:** $(x_1, y_1) \ldots (x_N, y_N), (x_w, y_w)$
**Ensure:** Salient Filter Ranking
1: $\bar{S} \leftarrow []$
2: **for** $i \leftarrow 1$ to $N$ **do**
3:     Calculate $\bar{\mathbf{s}}_i$ for $(x_i, y_i)$ by Eq. 1
4:     $\bar{S}$.append($\bar{\mathbf{s}}_i$)
5: **end for**
6: Estimate $\boldsymbol{\alpha}, \boldsymbol{\beta}$ from $\bar{S}$ using (Siffer et al., 2017)
7: Calculate $\bar{\mathbf{s}}_w$ for $(x_w, y_w)$ by Eq. 1
8: Calculate $\mathbf{p}_w$ for $\bar{\mathbf{s}}_w$ using $(\boldsymbol{\alpha}, \boldsymbol{\beta})$ and Eq. 5
9: salient_filter_ranking $\leftarrow$ argsort($\mathbf{p}_w$)

Figure 2: Detecting Salient Filters with POT

Figure 2 shows our salient filter ranking algorithm with POT. Let $\mathbb{R}$ be a real space and $L$ be the total number of convolution filters. The bold variables in Fig. 2 are all $L$-dimensional real vectors, i.e., $\bar{\mathbf{s}}_i, \boldsymbol{\alpha}, \boldsymbol{\beta}, \mathbf{p} \in \mathbb{R}^L$, where the $j$-th element in each vector is the value corresponding to the $j$-th convolution filter. Denote by $N$ the number of images in the validation set in the dataset. First, we calculate the saliency profiles for convolution filters, $\bar{\mathbf{s}}_j$ $(j = 1, \ldots, N)$, according to Eq. 1 for each image in the validation set, $(x_1, y_1) \ldots (x_N, y_N)$. Then, we perform the maximum likelihood estimation of the GPD parameters, $\boldsymbol{\alpha}$ and $\boldsymbol{\beta}$ in Eq. 4 , using the profiles $(\bar{\mathbf{s}}_1, \ldots, \bar{\mathbf{s}}_N)$. When a misclassified input is discovered, where $x_w$ and $y_w$ denote the misclassified input and its true label, we calculate the saliency profile of each filter for $x_w$ and store the saliency profiles that exceed the corresponding threshold. Then we compute the probability according to Eq. 5 for the filters with their saliency profile above the threshold. Finally, we can obtain the desired ranking by sorting these probabilities in ascending order. For the flowchart of the algorithm, please refer to the Fig. 6 in the Appendix. We used the maximum likelihood estimation of GPD parameters from the work by (Siffer et al., 2017) for our implementation and the threshold for each filter was set to the 90-th percentile value of the observed saliency profiles on the validation set of the dataset.

## 5 EXPERIMENT

In this section, we empirically analyzed the differences between POT-sailency and existing methods. In particular, we investigated what biases might occur and what problems the existing method might cause.

Levin et al. (2021) proposed two evaluation methods for quantitatively measuring the effect of detected filters: pruning and fine-tuning. For the pruning-based evaluation, we set all values in the filter to zero instead of actually modifying the model architectures. These removed filters will no longer affect the classification result because convolution is performed through the sum of the Hadamard product between the window of an input and the convolutional filters. In contrast, in the fine-tuning-based evaluation, we update the salient filters where it is assumed that if we correctly identify the filters causing misclassification, fine-tuning them would improve performance. For these experiments, we used the pretrained ResNet-50 provided by PyTorch framework as in Levin et al. (2021). The results of VGG and ViT are also shown in the Appendix.

### 5.1 EMPIRICAL ANALYSIS IN IMAGENET

We analyzed the original saliency and POT-saliency ranking methods in terms of two evaluation methods. We applied them to the ImageNet validation set. This experiment follows the one in Levin et al. (2021).

#### 5.1.1 FILTER-PRUNING-BASED EVALUATION

Starting from the top of the ranking of filters, we gradually turned off the filter and measured model performance according to the metrics by pruning up to $50$ filters. After all the incorrectly classified samples in the ImageNet validation set were processed, we average the results and the values are reported for each metric. We made comparisons among the original saliency ranking method, the POT-saliency ranking method, and a random-selection method in which we randomly chosed the convolution filters.

As shown in Fig. 3, both the original and POT-saliency methods share the same tendency to reduce the incorrect class confidence and increase correct class confidence and have almost the same ability to identify salient filters. Incorrect class confidence dropped by $25\%$ when $50$ salient filters were turned off, although choosing random filters did not decrease the confidence much. Also, we observed that the correct class confidence rose faster when salient filters were eliminated. These results suggest that the wrong classification is more or less due to these salient filters. The percentage of corrected samples for the POT-saliency method is higher than the random method as well, reaching $12\%$.

It is worth noting that zeroing out random filters helps the model classify well, and there are a couple of possible reasons for this. First, randomly chosen filters can include salient filters and these salient filters have an influence on the output. this hypothesis is consistent with the experiment conducted

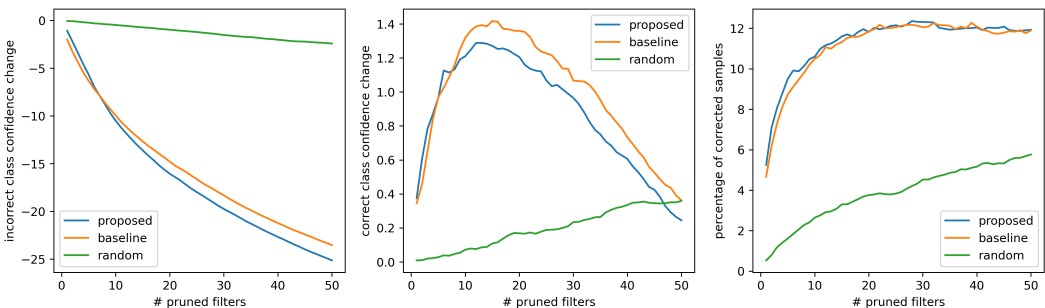

Figure 3: Comparisons on metrics among three methods. The POT method and baseline method have show the ability to drop incorrect class confidence.

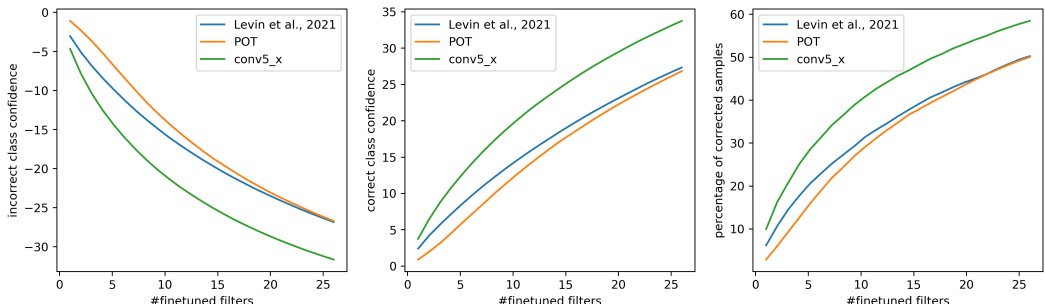

Figure 4: The POT method and baseline method have almost the same ability to detect malfunctioning filters in terms of one step fine-tuning.

by Levin et al. (2021), where neither correct nor incorrect class confidence changed when the least salient filters found by the baseline method were removed. Second, convolutional filters whose values are set to 0 could begin to equally contribute to the output for all the classes, leading to more evenly distributed confidence.

### 5.1.2 FILTER FINE-TUNING EVALUATION

We performed one step fine-tuning of ResNet-50 and observed the behavior change. For one step fine-tuning, we set the learning rate to 0.001 and multiplied this value to the gradients and subtracted the values from the original parameter values. We used the same metrics to measure the effect as in the filter pruning. One step fine-tuning may seem odd at first sight; however, we argue that fine-tuning for one step has several advantages over usual fine-tuning. Firstly, salient filter ranking will change after the modification of model parameters. Once the parameters in a filter are updated, the distributions of the gradient magnitude will be different. This forces us to compute the parameters for new GPDs and redo the whole process again. Therefore, one step fine-tuning can reduce the computation and is more practical. Secondly, the use of one step fine-tuning provides greater flexibility in selecting the number of filters for the model, thus enabling us to find the best configuration. After we compute the gradients and save them, we can easily increase or decrease the number of fine-tuned filters, because each parameter can be expressed by $\theta_i$ or $\theta_i - \lambda \nabla_{\theta_i} \mathcal{L}(f_\theta, x, y)$, where $\lambda$ is the learning rate. In contrast, if we perform normal fine-tuning, which needs several update operations, the gradient after the first update is dependent on the number of fine-tuned filters, and therefore we would need to start fine-tuning from scratch if we want to change the number of fine-tuned filters.

We conducted the experiment on the ImageNet validation set using the same GPD parameters computed previously for each filter, and compared our method to the baseline method. Figure 4 shows the result for the original and POT-saliency methods on the ImageNet validation set. It is clear that both methods transfer the confidence in the originally misclassified class to that of the correct class. In addition, we can see that half of the misclassified images is correctly classified after performing one step fine-tuning to 25 filters.

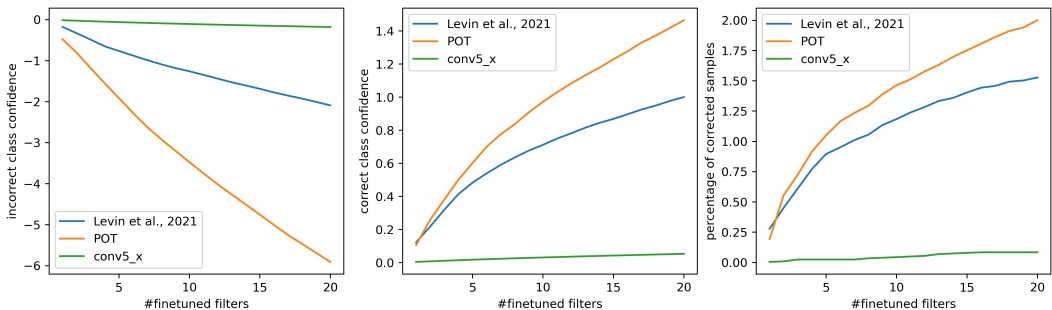

Figure 5: Comparison among three methods when ResNet-18 is trained using MNIST and used on SVHN. We evaluate them with the same metrics as before.

### 5.1.3 DISCUSSION: IS IT REASONABLE TO EVALUATE RANKING USING IMAGENET?

Considering the results of the experiments in the previous section, we can see that original and POT-saliency ranking successfully detects the filters inducing misclassification and that modifying these filters works positively for each input. However, we can also guess that manipulating parameters in the latter part of the convolutional layers is more likely to yield better changes in the results compared with manipulating the parameters of the former part. In fact, Kirichenko et al. (2022) showed that retraining the last layer can help ImageNet-trained models perform well on spurious correlation benchmarks. Although the last layer of CNNs is a linear layer, we presume that the same phenomenon would occur if we retrain the filters that belong to the convolutional layers in the latter half.

The ResNet architecture consists of five groups of convolutional layers: conv1, conv2_x, conv3_x, conv4_x, and conv5_x(He et al., 2016). The numbers of filters in these groups are listed in the following table. From tab. 3 in the appendix, we decided to focus on conv5_x, which contains nearly half of the filters. We constructed a simple algorithm: when a model makes a misclassification, we choose filters with higher gradient from conv5_x and perform one-step fine-tuning on them.

To evaluate the conv5_x fine-tuning approach, we conducted an experiment using the same setup as in sec. 5.1.2. Interestingly, the performance after fine-tuning filters in conv5_x outperforms both the original and POT-saliency methods by 5 to 10% for all the metrics as shown in Fig. 4. For this result, we hypothesized that, when training on ImageNet, which is huge in size and consists of a wide variety of classes, useful feature extractors are learned, so that fine-tuning can be reconciled by simply fine-tuning the the filters in conv_5, rather than by the filters in the feature extractors. Even if some filters in the feature extractor actually need to be modified, it is not possible in this situation to clarify whether they have been found. Therefore, in the next section, we propose evaluating each method in the domain shift problem setting, where the feature extractor filters clearly need to be modified.

### 5.2 EMPIRICAL ANALYSIS IN DOMAIN SHIFT

To find out whether conv5_x fine-tuning can operate well under any condition, we used datasets that show domain shift to ensure that CNNs as a feature extractor can only perform poorly. Domain shift is a common challenge in machine learning when the source domain and the target domain differ significantly. There are multiple possible triggers that give rise to the problem, one of which is the different feature space for the source and target domains. For example, we can intuitively understand that most models trained only with the MNIST dataset cannot extract useful features when they are used on the SVHN dataset as shown in Fig. 7 in Appendix. Since convolutional layers are involved in different types of feature extraction(Zeiler & Fergus, 2014), we expect the cause of misclassifications to be distributed among various convolutional layers.

We conducted our experiments with the MNIST and SVHN datasets. We trained ResNet-18 from scratch with MNIST dataset and applied our method to the SVHN training set to approximate the distributions of filter saliency. Then we analyzed where the top ranking filters are from. We did not use the pretrained model so as to avoid using models that already have a decent feature extractor.

Figure 5 shows how the performance changed on incorrectly classified images after fine-tuning. As we can see, modifying the filters in conv5_x only was not effective at all, changing almost nothing across all of the metrics. Interestingly, the POT method showed a better performance than the baseline method, which is a different result from that in Fig. 4. Especially, the POT method decreased the incorrect class confidence by up to 6%, whereas the baseline achieves only 2%, showing superior capability of discovering filters that contribute to misclassifications.

### 5.3 BIAS BEHIND SALIENT FILTERS: WHAT CAUSES PERFORMANCE DIFFERENCE BETWEEN POT-SALIENCY AND ORIGINAL SALIENCY?

In previous section, we explored the performance difference among various approaches. Now, we want to figure out where it comes from. For this purpose, we analyze the distribution of the chosen filters. More specifically, we counted how many times each filter ranked in the top 20 or 25, aggregated the results within the five groups, and calculated the proportion to clarify the general trends. As we can see from the results in Tables 1 and 2, POT-saliency ranking chose filters that belong to a wide range of layers, while original saliency ranking mainly chose conv5_x filters. This indicates that our method successfully reveals the fact that the model trained with MNIST is not capturing important features.

Table 1: Rate (%) of top-20 salient filters of ResNet-18 from each group on SVHN.

|  | baseline | POT |
|---|---|---|
| conv1 | 0 | 7.5 |
| conv2_x | 0.5 | 2.3 |
| conv3_x | 1.0 | 19.8 |
| conv4_x | 2.5 | 31.0 |
| conv5_x | 96.0 | 40.0 |

Table 2: Rate (%) of top-25 salient filters of ResNet-50 from each group on ImageNet.

|  | baseline | POT |
|---|---|---|
| conv1 | 0.1 | 7.2 |
| conv2_x | 0.6 | 24.8 |
| conv3_x | 1.5 | 2.1 |
| conv4_x | 24.2 | 15.1 |
| conv5_x | 73.6 | 50.8 |

These findings suggest that the baseline method is biased to choose filters from later groups such as conv5_x. To illuminate what causes this bias, we go back to the original saliency ranking method itself. The score for ranking generation is computed by performing the z-score normalization to parameter saliency, or filter saliency in the case of CNNs. Since ResNet adopts ReLU as an activation function, the gradient accumulates and grows bigger during the course of backpropagation unless the norm of weights is restricted to be small. Thus, the mean gradient is larger for conv1 and smaller for conv5_x, and the larger the mean value, the more likely it is to increase the standard deviation. In fact, as Fig. 10 in the appendix shows, the mean and std of the gradient gradually decreases from conv1 to conv5_x for ResNet-50. We divide the saliency profile by std when calculating the score, and this operation is presumably what introduces the bias.

## 6 CONCLUSION

We explored the parameter saliency for a CNN through the lens of EVT and provided POT-saliency. We analyzed the property of the original and POT-saliency ranking methods and found that the POT-saliency ranking method chooses from a wide range of convolutional layers while the baseline method has a bias to choose from the later part of the layers. We believe that this novel application of EVT in deep learning has the potential to open up new fields.

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

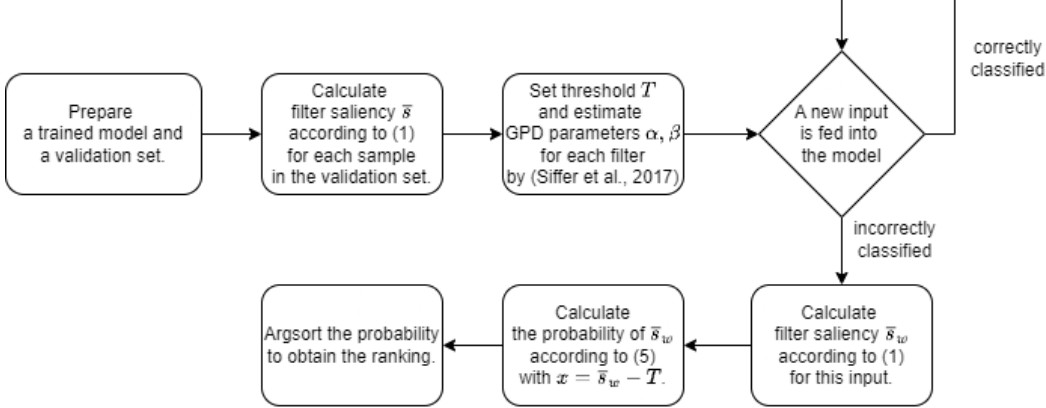

Figure 6: A flowchart illustrating how POT-saliency method works.

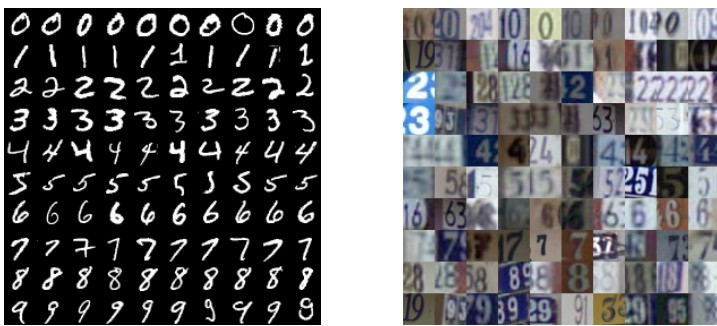

Figure 7: Visualization of digits in MNIST dataset (left) and SVHN dataset (right). SVHN digits come with colors, diverse computer fonts and various background from streets, while MNIST digits have black background only.

## A ARCHITECTURAL DIFFERENCES

We performed one-step fine-tuning for multiple CNN architectures such as VGG19. For this task, we set the learning rate to be 0.001 just as Sec. 5.1.2. The results are shown below. Figure 8 tells us that our method is effective for not only the ResNet architecture, but also other CNN architectures.

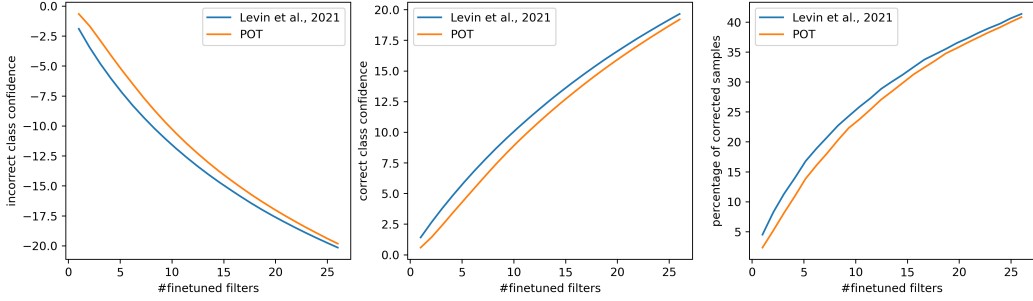

Figure 8: Comparisons on metrics for VGG19.

We also conducted this experiment on Vision Transformer. Since its architecture fundamentally differs from CNN, we needed to determine parameters corresponding to convolutional filters. We considered the parameters $W_k, W_q$, and $W_v$, which perform linear transformations to the input to obtain vectors of query, key, and value, to be particularly relevant to feature extraction. We treated the average magnitude of gradients for these parameters as parameter saliency. Similar to previous

Table 3: The number of filters in each group of convolutional layers for ResNet architecture.

| model name | conv1 | conv2_x | conv3_x | conv4_x | conv5_x |
|---|---|---|---|---|---|
| ResNet-18 | 64 | 256 | 512 | 1024 | 2048 |
| ResNet-50 | 64 | 1152 | 3072 | 9216 | 9216 |

experiments, we conducted experiments using the ImageNet validation set and used a learning rate of 0.001. The architecture is ViT-B-16. The results shown in Fig. 9 tell us that the POT-saliency method is not limited to CNNs only.

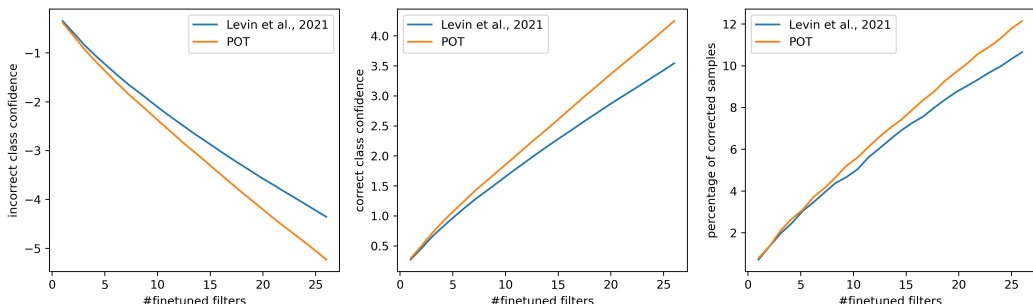

Figure 9: Comparisons on metrics for Vision Transformer.

## B  PROOF OF PROPOSITION 1

Since we have $\bar{S}_j \sim \mathcal{N}(\mu_j, \sigma_j)$, $\hat{S}_j$ follows the standard normal distribution:

$$\hat{S}_j \sim \mathcal{N}(0, 1). \tag{7}$$

The standard normal distribution has the following complementary error function $\mathrm{erfc} : \mathbb{R} \to [0, 1]$:

$$\mathrm{erfc}(x) = \frac{2}{\sqrt{\pi}} \int_x^\infty e^{-t^2} dt, \tag{8}$$

and we have $\mathbb{P}(\hat{S}_j > x) = \mathrm{erfc}(x)$ for any $j$. The derivative of the complementary error function with respect to x is

$$\frac{d\,\mathrm{erfc}}{dx}(x) = -\frac{2}{\sqrt{\pi}} e^{-x^2}. \tag{9}$$

The function $\mathrm{erfc}(x)$ is monotonically decreasing, and therefore if $\hat{s}_j \leq \hat{s}_{j'}$ holds true, we also have $\mathbb{P}(\hat{S}_j > \hat{s}_j) \leq \mathbb{P}(\hat{S}_{j'} > \hat{s}_{j'})$ and vice versa. Lastly, we have

$$\mathbb{P}(\hat{S} > \hat{s}) = \mathbb{P}\left(\frac{\bar{S} - \mu}{\sigma} > \frac{\bar{s} - \mu}{\sigma}\right) = \mathbb{P}(\bar{S} > \bar{s}), \tag{10}$$

and this concludes the proof.

## C  TUTORIAL FOR EXTREME VALUE THEORY

For those who are unfamiliar with the extreme value theory, we give a summary of (a) the motivation, (b) two main theorems, and (c) some methods to model extreme values. It would be especially helpful to understand the assumption necessary for the POT method.

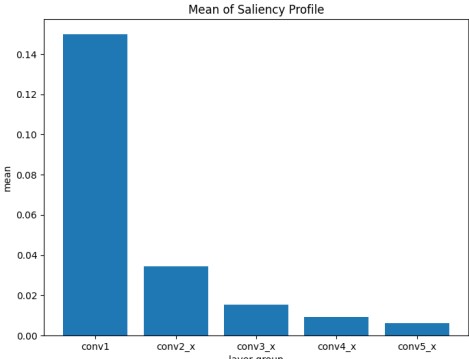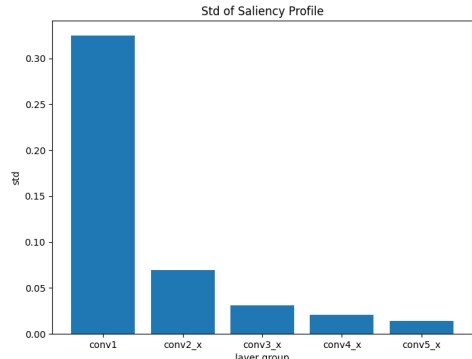

Figure 10: The mean and std of parameter saliency for ResNet-50 on ImageNet validation set.

### C.1 TWO MAIN THEOREMS

As described before, we try to model the maximum value of sequential data $M_n = \max\{X_1, X_2, ..., X_n\}$, where the data points are i.i.d.. To measure the potential value of it, we want to know the probability $\mathbb{P}(M_n \leq z)$ for some large value $z \in \mathbb{R}$. The value can be transformed with the distribution function $F$ as follows:

$$
\begin{aligned}
\mathbb{P}(M_n \leq z) &= \mathbb{P}(X_1 \leq z) \times \cdots \times \mathbb{P}(X_n \leq z) \\
&= \{F(z)\}^n.
\end{aligned}
\tag{11}
$$

If the distribution $F$ is well-known and can be expressed by an equation, we achieve our goal. However, in most of the case, this is not true. Thus, we need to approximate $F^n$. One problem arising here is that as $n$ increases, $\{F(z)\}^n$ converges 0 if $z < \tau$, where $\tau$ is the right end point of the distribution $F$. Avoiding this problem requires the linear normalization with two sequences of constants $\{a_n > 0\}$ and $\{b_n\}$:

$$
M_n^* = \frac{M_n - b_n}{a_n},
\tag{12}
$$

and we analyze $M_n^*$ instead.

The first theorem handles possible distributions for $\mathbb{P}(M_n^* \leq z)$ to converge as $n \to \infty$.

**Theorem 2.** *If there exists $\{a_n > 0\}$ and $\{b_n\}$ such that $\mathbb{P}\left(\frac{M_n - b_n}{a_n} \leq z\right)$ converges in law to $G(z)$ as $n \to \infty$, Then $G$ is one of the following three distribution functions:*

*1. $G(z) = \exp\left\{-\exp\left[-\left(\frac{z-b}{a}\right)\right]\right\};$*

*2. $G(z) = \begin{cases} 0 & z \leq b, \\ \exp\left\{-\left(\frac{z-b}{a}\right)^{-\alpha}\right\} & z > b; \end{cases}$*

*3. $G(z) = \begin{cases} \exp\left\{-\left[-\left(\frac{z-b}{a}\right)^{\alpha}\right]\right\} & z < b, \\ 1 & z \geq b, \end{cases}$*

*where $a > 0$ and $\alpha > 0$.*

These distributions are the families of Gumbel, Fréchet, and Weibull distributions. The three classes of distributions are together called extreme value distributions (EVD). These families are known to be represented uniformly and is called generalized extreme value (GEV) distributioin:

$$
G(z) = exp\left\{-\left[1 + \xi\left(\frac{z-\mu}{\sigma}\right)\right]^{-1/\xi}\right\},
\tag{13}
$$

where $z$ is constrained on $1 + \xi \left( \frac{z-\mu}{\sigma} \right) > 0$. The Gumbel distribution corresponds to $\xi \to 0$, and $\xi > 0$ and $\xi < 0$ are the cases for the Fréchet, and the Weibull distributions, respectively.

Now, we know that the unknown distribution for $M_n^*$ can be approximated by a GEV distribution, so we have reduced the approximation problem to the parameter estimation problem. We originally want to know the behavior of $\max M_n$, but we can achieve this if we can estimate parameters because $M_n$ also follows the GEV distribution:

$$\mathbb{P}(M_n \le z) = \mathbb{P}(M_n^* \le (z - b_n)/a_n) \approx G((z - b_n)/a_n). \tag{14}$$

One popular method is called the block maxima method, where we divide a sequence of data into several blocks, obtain the maximum value in each block, and fits the GEV distribution. we do not introduce the parameter estimation techniques because it is beyond scope of this paper, but many approaches, such as maximum likelihood estimation, have been developed so far.

A shortcoming of block maxima is that the method does not necessarily use extreme values, which are inherently scarce. The threshold-based method arises to avoid such a problem. In this case, values that exceeded a high threshold $T$ are considered to be extreme. The target of analysis changes to the probability:

$$\mathbb{P}\{X > T + x | X > T\} = \frac{1 - F(T + x)}{1 - F(T)}, \tag{15}$$

where $x > 0$ is the exceedance from the threshold. Again, we deals with the unknown distribution $F$. The second main theorem is Thm. 1 as described in Sec. 3.2, but we can now state the theorem more clearly with Thm. 2.

**Theorem 3.** *Let $X_1, X_2, ...$ be a sequence of i.i.d. random variables with the same distribution F, and $M_n = \max \{X_1, ..., X_n\}$. Suppose F satisfies Thm. 2:*

$$\mathbb{P}(M_n \le z) = exp\left\{ -\left[ 1 + \xi \left( \frac{z - \mu}{\sigma} \right) \right]^{-1/\xi} \right\},$$

*for some $\mu, \sigma > 0$ and $\xi$.*
*Then, for sufficiently large T, we have*

$$H(x) = 1 - \left( 1 + \frac{\xi x}{\hat{\sigma}} \right)^{-\frac{1}{\xi}}, \tag{16}$$

*where $x \in \{x > 0 \, and \, (1 + \xi x/\hat{\sigma}) > 0\}$ and $\hat{\sigma} = \sigma + \xi(T - \mu)$.*

The distribution of Eq. 16 is called the generalized pareto distribution. We can observe the close relationship between the GEV distribution and the generalized pareto distribution: they share the same $\xi$, and $\hat{\sigma}$ can be calculated from $\mu$ and $\sigma$ of the GEV distribution.

An outline proof is the following.

*Proof.* From the assumption, we have

$$\mathbb{P}(M_n \le z) = \{F(z)\}^n = exp\left\{ -\left[ 1 + \xi \left( \frac{z - \mu}{\sigma} \right) \right]^{-1/\xi} \right\}. \tag{17}$$

Taking the logarithm of the right equation gives

$$n \log F(z) \approx -\left[ 1 + \xi \left( \frac{z - \mu}{\sigma} \right) \right]^{-1/\xi}. \tag{18}$$

Considering Taylor expansion around $F(z) = 1$, where $z$ is large enough, we obtain the equation $\log F(z) \approx -\{1 - F(z)\}$. Substitution into Eq. 18 yields

$$1 - F(z) \approx \frac{1}{n} \left[ 1 + \xi \left( \frac{z - \mu}{\sigma} \right) \right]^{-1/\xi}. \tag{19}$$

Finally, from Eq. 15 and Eq. 19, we obtain

$$
\begin{aligned}
\mathbb{P}\{X > T + x | X > T\} &\approx \frac{\left[1 + \xi\left(\frac{T + x - \mu}{\sigma}\right)\right]^{-1/\xi}}{\left[1 + \xi\left(\frac{T - \mu}{\sigma}\right)\right]^{-1/\xi}} \\
&= \left\{\frac{1 + \xi\left(\frac{T - \mu}{\sigma}\right) + \xi\frac{x}{\sigma}}{1 + \xi\left(\frac{T - \mu}{\sigma}\right)}\right\}^{-1/\xi} \\
&= \left[1 + \frac{\xi x}{\hat{\sigma}}\right]^{-1/\xi},
\end{aligned}
\tag{20}
$$

where $\hat{\sigma} = \sigma + \xi(T - \mu)$. $\qquad\square$

## C.2 MAXIMUM LIKELIHOOD ESTIMATION

we have described the connection between the two main theorems. In this section, we will explain how we can estimate the parameters for a generalized pareto distribution with maximum likelihood estimation. The probability density function for it is in the form:

$$
f(x | \xi, \hat{\sigma}) = \frac{1}{\hat{\sigma}}\left(1 + \frac{\xi x}{\hat{\sigma}}\right)^{-\left(1 + \frac{1}{\xi}\right)}.
\tag{21}
$$

The likelihood function is defined as $L(\xi, \hat{\sigma}) = \Pi_{i=1}^{N_t} f(x_i | \xi, \hat{\sigma})$, where $N_t$ is the number of data points that exceed the threshold. Therefore, we want to maximize

$$
\log L(\xi, \hat{\sigma}) = -N_t \log \hat{\sigma} - \left(1 + \frac{1}{\xi}\right) \sum_{i=1}^{N_t} \log\left(1 + \frac{\xi x_i}{\hat{\sigma}}\right).
\tag{22}
$$

For the maximum likelihood estimation, it is necessary to have its partial derivatives with respect to $\xi$ and $\hat{\sigma}$ equal to 0:

$$
\frac{\partial L}{\partial \hat{\sigma}} = 0,
\tag{23}
$$

$$
\frac{\partial L}{\partial \xi} = 0.
\tag{24}
$$

Solving this problem requires numerical optimization, so the answers can be numerically unstable. One technique called the Grimshaw's trick mitigates this problem.

$$
\begin{aligned}
\frac{\partial L}{\partial \hat{\sigma}} &= -\frac{N_t}{\hat{\sigma}} - \left(1 + \frac{1}{\xi}\right) \sum_{i=1}^{N_t} \frac{1}{1 + \xi x_i/\hat{\sigma}}\left(-\frac{\xi x_i}{\hat{\sigma}^2}\right) \\
&= -\frac{N_t}{\hat{\sigma}} + \left(1 + \frac{1}{\xi}\right)\frac{1}{\hat{\sigma}} \sum_{i=1}^{N_t} \frac{\xi x_i/\hat{\sigma} + 1 - 1}{1 + \xi x_i/\hat{\sigma}} \\
&= \frac{N_t}{\xi\hat{\sigma}} - \left(1 + \frac{1}{\xi}\right)\frac{1}{\hat{\sigma}} \sum_{i=1}^{N_t} \frac{1}{1 + \xi x_i/\hat{\sigma}} = 0;
\end{aligned}
\tag{25}
$$

$$
\begin{aligned}
\frac{\partial L}{\partial \xi} &= \frac{1}{\xi^2} \sum_{i=1}^{N_t} \log\left(1 + \frac{\xi x_i}{\hat{\sigma}}\right) - \left(1 + \frac{1}{\xi}\right) \sum_{i=1}^{N_t} \frac{x_i/\hat{\sigma}}{1 + \xi x_i/\hat{\sigma}} \\
&= \frac{1}{\xi^2} \sum_{i=1}^{N_t} \log\left(1 + \frac{\xi x_i}{\hat{\sigma}}\right) - \left(1 + \frac{1}{\xi}\right)\frac{1}{\xi} \sum_{i=1}^{N_t} \frac{\xi x_i/\hat{\sigma} + 1 - 1}{1 + \xi x_i/\hat{\sigma}} \\
&= \frac{1}{\xi^2} \sum_{i=1}^{N_t} \log\left(1 + \frac{\xi x_i}{\hat{\sigma}}\right) - \left(1 + \frac{1}{\xi}\right)\frac{N_t}{\xi} + \left(1 + \frac{1}{\xi}\right)\frac{1}{\xi} \sum_{i=1}^{N_t} \frac{1}{1 + \xi x_i/\hat{\sigma}} = 0.
\end{aligned}
\tag{26}
$$

Multiplying $\hat{\sigma}$ to Eq. 25 and $\xi$ to Eq. 26 yields

$$\frac{N_t}{\xi} - \left(1 + \frac{1}{\xi}\right) \sum_{i=1}^{N_t} \frac{1}{1 + \xi x_i/\hat{\sigma}} = 0, \tag{27}$$

$$\frac{1}{\xi} \sum_{i=1}^{N_t} \log \left(1 + \frac{\xi x_i}{\hat{\sigma}}\right) - N_t - \frac{N_t}{\xi} + \left(1 + \frac{1}{\xi}\right) \sum_{i=1}^{N_t} \frac{1}{1 + \xi x_i/\hat{\sigma}} = 0. \tag{28}$$

Adding the two equations gives:

$$\xi = \frac{1}{N_t} \sum_{i=1}^{N_t} \log \left(1 + \frac{\xi x_i}{\hat{\sigma}}\right). \tag{29}$$

Here we use the change of variables by the equation $\theta = \frac{\xi}{\hat{\sigma}}$. Note that both $\xi$ and $\hat{\sigma}$ can be calculated through:

$$\xi = \frac{1}{N_t} \sum_{i=1}^{N_t} \log \left(1 + \theta x_i\right), \tag{30}$$

$$\hat{\sigma} = \frac{\xi}{\theta}. \tag{31}$$

Therefore, our goal becomes finding the optimal $\theta$. The first step is to transform the log of likelihood function $\log L(\xi, \hat{\sigma})$ with $\theta$:

$$\log L(\xi, \hat{\sigma}) = -N_t \log \hat{\sigma} - \left(1 + \frac{1}{\xi}\right) \sum_{i=1}^{N_t} \log \left(1 + \frac{\xi x_i}{\hat{\sigma}}\right)$$

$$= -N_t \log \frac{\xi}{\theta} - \left(1 + \frac{1}{\xi}\right) N_t \xi \qquad (\because Eq.\ 30)$$

$$= -N_t \log \left(\frac{1}{\theta} \frac{1}{N_t} \sum_{i=1}^{N_t} \log \left(1 + \theta x_i\right)\right) - \sum_{i=1}^{N_t} \log \left(1 + \theta x_i\right) - N_t. \tag{32}$$

The derivative of this function with respect to $\theta$ is:

$$\frac{dL}{d\theta} = -N_t \frac{-\frac{1}{\theta^2} \frac{1}{N_t} \sum_{i=1}^{N_t} \log \left(1 + \theta x_i\right) + \frac{1}{\theta} \frac{1}{N_t} \sum_{i=1}^{N_t} \frac{x_i}{1+\theta x_i}}{\frac{1}{\theta} \frac{1}{N_t} \sum_{i=1}^{N_t} \log \left(1 + \theta x_i\right)} - \sum_{i=1}^{N_t} \frac{x_i}{1 + \theta x_i}$$

$$= \frac{N_t}{\theta} - N_t \frac{\sum_{i=1}^{N_t} \frac{1}{\theta} \frac{\theta x_i + 1 - 1}{1+\theta x_i}}{\sum_{i=1}^{N_t} \log \left(1 + \theta x_i\right)} - \sum_{i=1}^{N_t} \frac{1}{\theta} \frac{\theta x_i + 1 - 1}{1 + \theta x_i}$$

$$= \frac{N_t}{\theta} - N_t \frac{\frac{N_t}{\theta} + \frac{1}{\theta} \sum_{i=1}^{N_t} \frac{1}{1+\theta x_i}}{\sum_{i=1}^{N_t} \log \left(1 + \theta x_i\right)} - \left(\frac{N_t}{\theta} - \frac{1}{\theta} \sum_{i=1}^{N_t} \frac{1}{1 + \theta x_i}\right)$$

$$= -\frac{N_t}{\theta} \frac{N_t + \sum_{i=1}^{N_t} \frac{1}{1+\theta x_i}}{\sum_{i=1}^{N_t} \log \left(1 + \theta x_i\right)} + \frac{1}{\theta} \sum_{i=1}^{N_t} \frac{1}{1 + \theta x_i}. \tag{33}$$

Here, we assume that $N_t$ is a positive number. Otherwise, we cannot estimate the probability distribution because no values exceed the threshold. The optimal $\theta$ should satisfy $\frac{dL}{d\theta} = 0$, which

yields:

$$-\frac{N_t}{\theta}\frac{N_t + \sum_{i=1}^{N_t}\frac{1}{1+\theta x_i}}{\sum_{i=1}^{N_t}\log\left(1+\theta x_i\right)} + \frac{1}{\theta}\sum_{i=1}^{N_t}\frac{1}{1+\theta x_i} = 0 \tag{34}$$

$$N_t + \sum_{i=1}^{N_t}\frac{1}{1+\theta x_i} - \frac{1}{N_t}\left(\sum_{i=1}^{N_t}\frac{1}{1+\theta x_i}\right)\left(\sum_{i=1}^{N_t}\log\left(1+\theta x_i\right)\right) = 0 \tag{35}$$

$$N_t - \left(\sum_{i=1}^{N_t}\frac{1}{1+\theta x_i}\right)\left(1 - \frac{1}{N_t}\sum_{i=1}^{N_t}\log\left(1+\theta x_i\right)\right) = 0 \tag{36}$$

$$\left(\frac{1}{N_t}\sum_{i=1}^{N_t}\frac{1}{1+\theta x_i}\right)\left(1 - \frac{1}{N_t}\sum_{i=1}^{N_t}\log\left(1+\theta x_i\right)\right) - 1 = 0. \tag{37}$$

Calculating the GPD parameters $\xi, \hat{\sigma}$ is now reduced to solving the Eq. 37. This is the full picture of the Grimshaw's trick. It can be shown that the equation has at least one solution other than $\theta = 0$ if the derivative of the left hand side of Eq. 37 equals to 0 at $\theta = 0$.

The lower bound for $\theta$ is given by

$$\theta > -\frac{1}{\max_i x_i}, \tag{38}$$

because $1 + \theta x_i$ should be positive for any $i$. Grimshaw shows the upper bound for $\theta$:

$$\theta < \frac{2(\bar{x} - \min_i x_i)}{(\min_i x_i)^2}, \tag{39}$$

where $\bar{x}$ is the mean of $x_1, \cdots, x_{N_t}$. Therefore, the implementation must find all the possible roots for the Eq. 37 and choose the optimal value according to $\log L(\theta)$.

## D    ADDITIONAL EXPERIMENTS

We conducted two additional domain-shift experiments. One is the ImageNet-C(Hendrycks & Dietterich, 2019) where various generated corruptions are applied to the ImageNet validation set. The other is PACS dataset. The PACS dataset consists of images from 7 classes, where P stands for photo, A for art, C for cartoons, and S for sketch. The dataset has 9,991 images in total.

### D.1    IMAGENET-C

For this experiment, we used the same pretrained ResNet50 as Sec. 5.1.2, and also the POT parameters are identical. The results are shown in Fig. 11. For the types of corruption, we chose gaussian noise, snow, pixelate, and contrast. In this scenario, We can see similar performance across all the corruptions. Since the ImageNet dataset is such a large dataset, the pretrained model is likely to have a good feature extractor. Thus, the bias of choosing filters from the latter part of convolutional layers behind (Levin et al., 2021)'s method has less influence.

### D.2    PACS DATASET

In addition to the MNIST and SVHN datasets, we also used the PACS dataset(Li et al., 2017) to examine the behavior of ResNet-18 when the filters were fine-tuned according to the POT-saliency method, the (Levin et al., 2021)'s method and the conv5_x method. We again started from non-pretrained ResNet18 and trained with 0.001 of learning rate with Adam. When training, we choose three source domains from P, A, C, and S and split images into training and validation set. Then, we used the remaining domain to conduct the evaluation. Henceforth, We will represent the domain used for evaluation in the last letter. For example, when we write PCSA, it means that we use art images for evaluation and train a model with photo, cartoon, and sketch images. The following four figures show the results.

In PACS and PASC, we can clearly see the difference between the POT method and (Levin et al., 2021)'s method, while PCSA and ACSP show the similar tendency. One possible reason for this

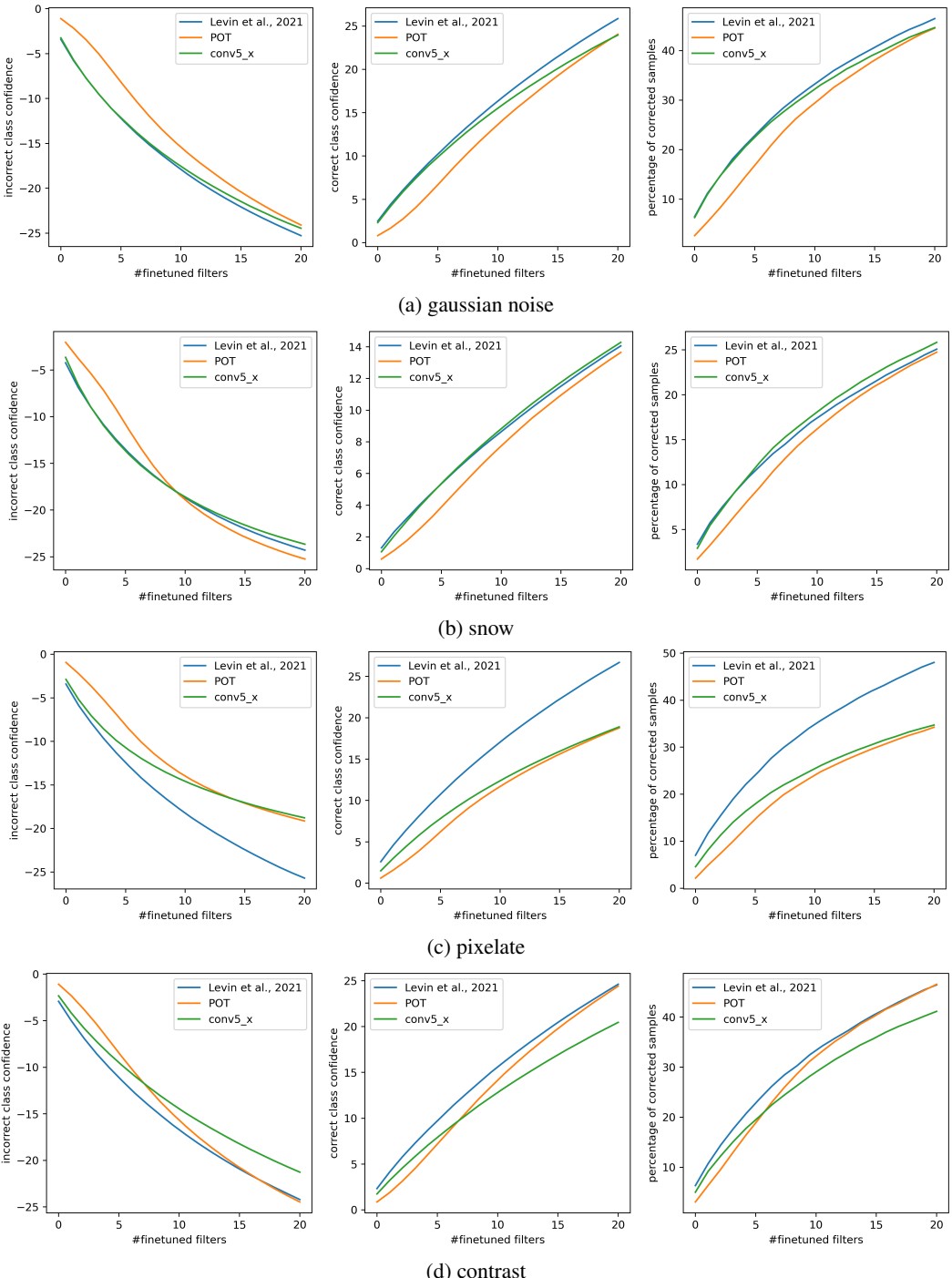

(a) gaussian noise

(b) snow

(c) pixelate

(d) contrast

Figure 11: The results of evaluation on ImageNet-c dataset.

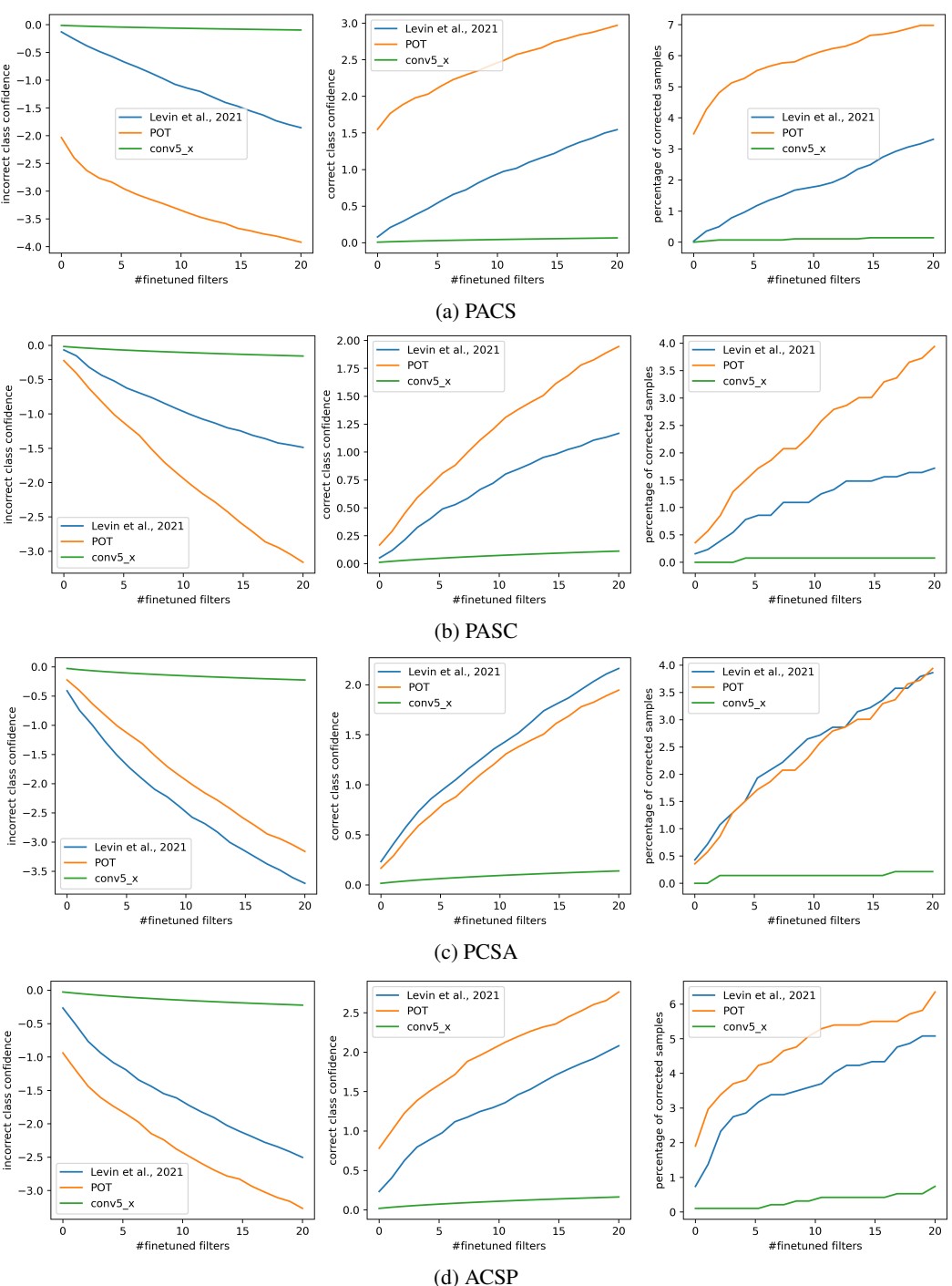

Figure 12: The results of evaluation on PACS permutation.

tendency can be found in the feature distribution of the PACS images in Fig. 13. We see photo and art painting images share the feature distribution, but cartoon and sketch images are independent and distant from the rest. When sketch and cartoon images are used for evaluation (i.e., PACS and PASC), the feature extractor must be changed so that they can successfully classify the images. In this case, the POT method shows less bias and outperforms the (Levin et al., 2021)'s method.

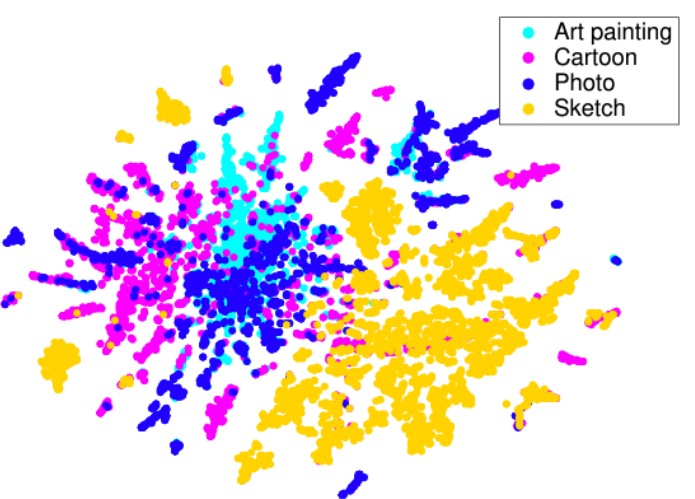

(c) The feature distribution of PACS

Figure 13: Distribution of features in PACS photos from (Li et al., 2017)
.

