# OpenReview forum: "Understanding Parameter Saliency via Extreme Value Theory"
_ICLR.cc/2024/Conference — Submitted to ICLR 2024_

### Official Review · Reviewer_HPzx · 2023-10-31

**Soundness:** 3 good
**Presentation:** 2 fair
**Contribution:** 3 good
**Rating:** 6
**Confidence:** 3

**Summary:**

This study falls into the field of parameter saliency in XAI. The draft proposes to apply the peaks-over-threshold (POT) technique on neural networks to locate salient parameters or filters. The POT is a solution in the Extreme Value Theory in statistics for anomaly detection.

**Strengths:**

This study explores a very important task in AI, explaining the black-box network. And it focuses on a very interesting topic, which parameters are salient or important in classification(misclassification).

**Weaknesses:**

Before discussing the weaknesses, I have to admit I have more expertise in CS, and my knowledge in statistics is limited. I might ask some naive questions and please correct me if I am wrong.

The structure of the draft can be better organized for a better readability, the current version is a little confusing. The overall idea is applying an existing method from statistics, POT, to locate salient parameters. The main baseline is (Levin 2021) which also studies the parameter saliency. However, the motivation of this study is not well introduced. In the introduction and the related work, a list of studies in XAI and statistics are sequentially displayed without discussing what kind of problems this draft can solve. For instance, Sec.2 "Related work", Interpretability, this paragraph is more like a basic tutorial, which is not directly related to this study, neither it is complete. The second paragraph in Sec2 is more about the CAM-based, which is not in the parameter space. I would suggest only discuss the most related parameter saliency, and POT related studies.

Preliminary 3.1 is re-introducing the study (Levin), which is not necessary. The most important part is missing in that section, whats the limitation in the previous solution(Levin) and how this solution can solve that. I can grab some scatter information from the draft, previous study assumes the gradient follows a normal distribution, and this assumption may not hold, as discussed in Motivation Sec 4.1. I cannot see a clear demonstration that POT can solve this problem or I might miss this. Prior to the solution, this draft didnot showcase the problem of the normal distribution assumption. I know most CS scientist naively assume normal distribution, because it is simple yet effective. If this assumption is problematic, they can simple assume a more specific distribution. Could the normal distribution be "simple" enough for the study now? From the view of the solution, Appendix C, EVT normally add more assumptions to catch more distributions, which means the solution could be either not necessary (the normal assumption is enough) or limited (could the combined EVD catch everything?) I might miss something in the paper, proposition 1 and appendix B also are based on normal distribution, can I ask why a simple normal distribution can be representative enough in this problem raised?

Without a clear problem introduction, the proposed method is not clear and convincing in enough at this moment. And the results in Figs 3 and 4 also show that the difference is tiny, which echos the question, do we really need a more complicated distribution for this purpose now?

Tiny suggestion, I know Figs 3 and 4 have different components, but it is better to use the same colour and the same name of the same thing, e.g., POT vs Levin, or (proposed vs baseline). I searched around the paper, has the term "baseline" been explained?

**Questions:**

Please see the question listed above, the current draft randomly grabbing a technique from statistics to CS without introducing the need. Thus the clarity and novelty could be further improved before publishing.

---

> ### Author Response · Authors · 2023-11-17
> **Response to Reviewer HPzx**
>
> We appreciate your careful considerations on improving the work!
>
> ### Questions and Answers
>
> > Q.1 Do we really need a more complicated distribution for this purpose now?
>
> A.1 We are actually further relaxing the assumption of a normal distribution. Although EVT may seem to require more assumptions, it is known that extreme values of many standard distributions follow this distribution, and the normal distribution is no exception. It is known that the maximum value from the normal distribution converges to the Gumbel distribution. It also has been demonstrated that EVT can handle various distributions, including exponential and uniform distributions. Therefore, there is no implicit need to assume a normal distribution, making it applicable to a broader range of distributions.
>
> > Q.2  What's the limitation in the previous solution(Levin) and how this solution can solve that?
>
> A.2 As stated in Section 4.1, assuming a normal distribution itself can be considered a limitation. This is particularly true when dealing with heavy-tailed distributions, where values like mean and variance are significantly influenced by outliers, making them unreliable as proper statistical measures. In the existing method (Levin et al.), the use of z-score transformation may be strongly affected by this, potentially being unable to achieve accurate comparisons. Moreover, comparing values calculated from statistics of distributions that are not necessarily the same may inherit the characteristics of each distribution. For instance, by examining the tables in Section 5.3, it becomes evident that the existing method struggles to select filters from the early convolutional layers. This is attributed to the high magnitude of gradients and increased variance in the early layers. In contrast, the POT method transforms the discussion into a probabilistic context, enabling a unified comparison irrespective of the underlying distribution.
>
>
> > Q.3  I might miss something in the paper, proposition 1 and appendix B also are based on normal distribution, can I ask why a simple normal distribution can be representative enough in this problem raised?
>
> A.3 Proposition 1 and Appendix B pertain to the methodology of Levin et al. In their approach, they employ z-score transformation as depicted in Equation (2). While z-score transformation implicitly assumes a normal distribution, Proposition 1 asserts that under such an assumption, the ranking score function can be viewed as a probability. By introducing a probabilistic perspective in this manner, a new method utilizes extreme value theory to enable statistically meaningful comparisons.
>
>
> ### Others
> - Thank you for pointing out the inconvenience of the legend and colors in the figure. We will fix it.
> - For the suggestion on the related work, parameter saliency is a new concept and related work is really scarce. Also, POT is rarely used in the context of machine learning. It is rather used in fields like hydrology, finance, and environmental science. So we consider XAI related works better fit the section.

---

> > ### Comment · Reviewer_HPzx · 2023-11-22
> > **response**
> >
> > I would like to thank the authors for the further explanation, which answers my main questions regarding the use case of EVT. The explanation of the limitation on normal distribution is clear to me now and the proposed method is based on a looser assumption. I would raise my rating.

---

> > > ### Author Response · Authors · 2023-11-22
> > > **Thank you!**
> > >
> > > Dear Reviewer HPzx
> > >
> > > Thank you for your thoughtful review. Your questions address one of the most important aspects of our paper. We also express our gratitude for raising the score.

---

### Official Review · Reviewer_nAUC · 2023-11-02

**Soundness:** 3 good
**Presentation:** 3 good
**Contribution:** 2 fair
**Rating:** 5
**Confidence:** 4

**Summary:**

This paper investigated the parameter saliency for a CNN through the lens of EVT and provided POT-saliency.
This paper also analyzed the property of the original and POT-saliency ranking methods and found that the
POT-saliency ranking method chooses from a wide range of convolutional layers while the baseline
method has a bias to choose from the later part of the layers.

**Strengths:**

originality - The proposed concept is interesting. The POT-salience method is expected to introduce semantic features to the framework POT and hence improve the classification performance.

quality - The paper presents theoretical and experimental results to demonstrate the proposed algorithm. Both theory and experiments are comprehensive.

clarity - The experimental section clearly shows that the proposed POT-salience algorithm performs better than the other methods in different settings.

significance - Upon the justified theory, the proposed method will have strong impacts on the standard CNN architecture.

**Weaknesses:**

originality - Very confusing explanations on the new proposal in this paper. For example, it is not clear how to link Theorem 1 in Eqs. 3 and 4 with the POT method?  More detailed discussion is used to demonstrate the novelty.

quality - The entire paper reads imbalanced due to the heavy weights on the theoretical justification.  The proposed algorithm lacks a full scale algorithmic discussion, including parameter update and initialization.

clarity - The current form of the paper is confusing in terms of the theoretical proof. There is no clear algorithmic flowchart and discussion. The experimental results are partial and lacks comparisons against different POT variants.

significance - The outcomes of the experiments are not significant, compared against those of the other state of the arts.

**Questions:**

1. How does the proposed POT algorithm work? A flowchart may be used to help the discussion.
2. More comparison experiments need to be conducted but the current version is weak in terms of experimental work.
3. How will the parameters in the proposed POT system affect the system performance?

---

> ### Author Response · Authors · 2023-11-17
> **Response to Reviewer nAUC**
>
> We appreciate your thoughtful feedback, which has been invaluable in improving the quality of our work.
>
> > Q.1 How does the proposed POT algorithm work? A flowchart may be used to help the discussion. ( how to link Theorem 1 in Eqs. 3 and 4 with the POT method?)
>
> A.1 Thank you for your suggestion. We have added a flowchart illustrating the operation of the POT algorithm in Figure 6 of the appendix. If you have any uncertainties, please feel free to inquire with us again.
>
> Regarding the relationship between Theorem 1 and the POT method, Theorem 1 means that the more you move towards the tail of the probability distribution, the better it can be approximated by the Generalized Pareto Distribution (GPD). The POT method is a technique for estimating the parameters of this GPD, utilizing extreme values exceeding a sufficiently large threshold for maximum likelihood estimation.
>
> > Q.2 More comparison experiments need to be conducted but the current version is weak in terms of experimental work.
>
> A.2 We will add some comparison experiments within the next few days, we would appreciate it if you could wait until then.
>
> > Q.3 How will the parameters in the proposed POT system affect the system performance?
>
> A.3 One critical parameter to consider for POT is the threshold. Generally, if the threshold is set too high, the amount of data available for estimating GPD parameters becomes limited, resulting in increased variance. On the other hand, if the threshold is set too low, the approximation accuracy of GPD tends to deteriorate.
>
> In terms of the POT method, it is worth noting that since the algorithm generates rankings for filters that exceed the threshold, the number of filters included in the ranking is influenced by the choice of the threshold. This is the largest difference between POT method and Levin et al.'s method because the latter one always gives us the ranking of the size of the total filter number.
>
> We did an experiment on ImageNet dataset by setting the threshold to 95-th percentile of gradient data, but it did not show much difference. This can be because the ImageNet dataset is very large and the threshold has a little impact on maximum likelihood estimation of POT parameters.
>
> > Q.4  The outcomes of the experiments are not significant, compared against those of the other state of the arts.
>
> A.4 We claim that the primary objective is to comprehend the existing research (Levin et al.) from the perspective of statistical anomaly as the title suggests. Although we compare several approaches in the experiments, parameter saliency is a concept proposed recently and there is a "knowledge gap in terms of understanding why parameter saliency ranking can find the filters inducing misidentification" as we stated in the introduction. Therefore, we do not consider it as weakness.

---

> > ### Author Response · Authors · 2023-11-20
> > **Response to Reviewer nAUC (2)**
> >
> > > Q.2 More comparison experiments need to be conducted but the current version is weak in terms of experimental work.
> >
> > We have added experiments on the ImageNet-C and PACS datasets as domain adaptation datasets. For ImageNet-C, we used the pretrained ResNet50 and the POT parameters estimated from the ImageNet dataset (which are the same as in the experiments in Sec.5.1.2). The results in Fig.11 show that we do not see much difference among the three methods, and conv5 is not as effective as the original ImageNet experiment (Fig.4). The ImageNet dataset contains many images and the pretrained model acquires a good feature extractor throughout the whole training process. In this case, the bias of choosing filters from the latter part of convolutional layers behind Levin et al.’s method is not much of a problem.
> >
> > We obtained more important findings from the PACS dataset. The PACS dataset consists of images from 7 classes, where P stands for photo, A for art, C for cartoons, and S for sketch. The dataset has 9,991 images in total. This time, we used non-pretrained ResNet18 and trained from scratch just as MNIST-SVHN experiment. When train a model, we choose three domains from P, A, C, and S and evaluate on the remaining domain. From Fig.12, we can observe that the POT method was especially better when the model is trained on PAC and evaluated on S, and trained on PAS and evaluated on C. This can be attributed to the feature distribution of sketch and cartoon image in Fig.13. Features of sketch and cartoon images are not much learnable from images from other domains, and it requires tuning the feature extractor. However, other methods show bias here, being unable to perform well.

---

### Official Review · Reviewer_8wd7 · 2023-11-04

**Soundness:** 2 fair
**Presentation:** 3 good
**Contribution:** 3 good
**Rating:** 6
**Confidence:** 2

**Summary:**

The paper investigates the topic of parameter saliency to understand misclassifications in neural networks. Unlike some prior works which focus on input saliency maps, parameter saliency is shown to be helpful to correct for misclassifications. Building over prior work, the paper formulates the ranking of salient filters through connections with statistical anomaly detection and extreme value theory. Empirical experiments show that the approach leads to plausible conclusions.

**Strengths:**

- As deep learning finds more and more applications in daily-life, interpretability is getting more important. The idea of focusing on parameter saliency is an interesting approach towards interpretability and model surgery.
- I am not an expert in the topic, but the paper is well written for the most part. Key ideas and intuitions are well explained.
- The approach is evaluated on multiple architectures and datasets.

**Weaknesses:**

- 5.1.2 : "We performed one step fine-tuning .. " -> Is the finetuning and validation being done on the same dataset ?
- The interpretations in Figure 4 are not clear, especially the conv5_x curve and its relation to the proposed approach.
- I think it will be more intuitive to think of %improvement in downstream performance rather than %corrected samples.
- How do you differentiate between label errors and misclassification ?
- A lot of things seem to have changed when moving to the domain adaptation experiments : architecture, dataset, model size, pretraining. Instead, it would be beneficial to show results on domain adaptation datasets like ImageNet-C, DomainNet etc.
- I am not too convinced with the one-step finetuning process apart from being "easy" to do.
- 4.3 : "anomalous behavior of filters.. is a rare event" - Can the authors give more insight into when would this hold ? Perhaps as a function of class difficulty, model size, dataset size.

**Questions:**

Please refer to questions in the weaknesses section.

[Minor] Writing suggestions:
- Fix tense : Abstract, ".. has efficiently corrected .. "
- Section 1 : "model's decision" ^making "process" ?
- Section 2 : "importance" -> "Importance"
- Reword title of Section 3.2
- Section 5 : "FOr the pruni .. "

---

> ### Author Response · Authors · 2023-11-17
> **Response to Reviewer 8wd7**
>
> Thank you sincerely for your thoughtful and constructive feedback on our paper.
>
> ### Questions and Answers
> >Q.1 Is the finetuning and validation being done on the same dataset ?
>
> A.1 In general, the dataset used for finetuning is distinct from the validation dataset. When using a pre-trained model provided by deep learning frameworks, there is no access to the actual validation set during training. Also ImageNet officially offers two datasets, “training set” and “validation set”, where people commonly split “training set” for training and validation. In our work and existing work(Levin et al.), “validation set” is used for calculating the statistics and performing one-step finetuning.
>
>
> >Q.2 The interpretations in Figure 4 are not clear, especially the conv5_x curve and its relation to the proposed approach.
>
> A.2 There is no specific relationship between conv5_x and POT. As described in section 5.1.3, Kirichenko et al. reported good performance when finetuning only the final layer. Additionally, in Tables 1 and 2 of Section 5.3, it can be observed that the approach of Levin et al. is heavily biased towards the latter convolutional layers. The conv5_x method is developed based on these observations. The conv5_x curve in Figure 4 demonstrates that the model performance improves by only finetuning the convolutional filters in the latter layers in descending order of gradient magnitude. This raises the question of whether it is necessary to modify feature extractor in the earlier layers when dealing with scenes requiring different features due to domain shift.
>
> >Q.3 I think it will be more intuitive to think of %improvement in downstream performance rather than %corrected samples.
>
> A.3 Our experimental metric aligns with the one used in the existing study (Levin et al). The objective of Levin et al.'s research is to identify which part of the model parameters is responsible for misclassifications. Investigating %corrected samples is effective because if correcting a limited number of filters through one-step finetuning leads to accurate classification, it validates the saliency of those filters.
>
> >Q.4 How do you differentiate between label errors and misclassification ?
>
> A.4 Our study is predicated on the occurrence of misclassifications. The primary objective is to comprehend the existing research (Levin et al.) from the perspective of statistical anomaly. The experiments and problem settings are conducted based on this research. Consequently, the analysis is carried out in an ideal scenario, assuming there are no factors such as label errors.
>
> >Q.5 I am not too convinced with the one-step finetuning process apart from being "easy" to do.
>
> A.5 This is a good point. We again note that one-step finetuning employs the same experimental methodology as the existing study (Levin et al.) and is utilized for evaluation.  We would like to examine changing the salient parameters can positively affect the classification result. Also, since the statistical measures, such as the average magnitude of gradients, would change with model parameters, conducting multiple steps of finetuning would be a more complex way of evaluation. Thus, one-step finetuning is used.
>
> >Q.6. the anomalous behavior of filters causing misclassification can also be considered a rare event" - Can the authors give more insight into when would this hold ? Perhaps as a function of class difficulty, model size, dataset size.
>
> A.6 In the context of gradients, the simplest explanation can be derived. As illustrated in Figure 1, there is a notably high frequency of instances where the magnitude of the gradient is nearly zero. This phenomenon occurs when the deep learning model correctly classifies the input. In other words, as the model's performance improves, instances where the gradient magnitude is significantly larger than zero become increasingly rare. Therefore, it is hypothesized that this becomes more achievable with larger dataset sizes and model sizes.
>
> ### Others
> - For experiments on domain adaptation datasets, we will add them within the next few days, so we would appreciate it if you could wait until then.
> - Thank you for writing suggestions. We addressed them and updated our draft.

---

> > ### Comment · Reviewer_8wd7 · 2023-11-19
> > **Thanks for the comments**
> >
> > Thanks for the comments. The rebuttal helps address some of my concerns.
> > - Q5 : I am not fully convinced about this yet. While I do understand the paper over which the paper builds upon uses this approach, this doesn't seem to be set in stone.
> > - Looking forward to the domain adaptation experiments.

---

> > > ### Author Response · Authors · 2023-11-20
> > > **Response to Reviewer 8wd7 (2)**
> > >
> > > > Q.5 : I am not fully convinced about this yet. While I do understand the paper over which the paper builds upon uses this approach, this doesn't seem to be set in stone.
> > >
> > > A.5 This approach is set in stone in parameter saliency analysis. We would like to emphasize that it is an evaluation method for parameter salicency analysis, which was first proposed in Levin et al. and no other evaluation method has been proposed so far, although one-step finetuning may sound like some way of finetuning a model. The primary purpose of this one-step finetuning evaluation is to validate that the salient parameters are the ones causing misclassification. In this evaluation, we need to identify the individual filters for each  misclassified image. Since we only consider updating parameters on a single sample at a time, it would have a positive or negative effect on the classification process of other samples, and it is not suitable for what we call “finetuning”. Levin et al. also say that this finetuning is not intended to be “a state-of-the-art fine-tuning”[1]. In Levin et al.’s work, the authors compare their method to randomly chosen filters and it is demonstrated that finetuning a few random filters does not impact the classification results as much as their method. So this evaluation way is just enough for the validation of the algorithm that finds salient parameters.
> > >
> > > [1]https://openreview.net/forum?id=qEGBB9YB31&referrer=%5Bthe%20profile%20of%20Roman%20Levin%5D(%2Fprofile%3Fid%3D~Roman_Levin1)#:~:text=not%20as%20a%20state%2Dof%2Dthe%2Dart%20fine%2Dtuning%20method
> > >
> > > ### Additional experiments
> > > Thank you for waiting for the additional experimental results. We have added experiments on the ImageNet-C and PACS datasets as domain adaptation datasets. For ImageNet-C, we used the pretrained ResNet50 and the POT parameters estimated from the ImageNet dataset (which are the same as in the experiments in Sec.5.1.2). The results in Fig.11 show that we do not see much difference among the three methods, and conv5 is not as effective as the original ImageNet experiment (Fig.4).
> > > The ImageNet dataset contains many images and the pretrained model acquires a good feature extractor throughout the whole training process. In this case, the bias of choosing filters from the latter part of convolutional layers behind Levin et al.’s method is not much of a problem.
> > >
> > > We obtained more important findings from the PACS dataset.
> > > The PACS dataset consists of images from 7 classes, where P stands for photo, A for art, C for cartoons, and S for sketch. The dataset has 9,991 images in total. This time, we used non-pretrained ResNet18 and trained from scratch just as MNIST-SVHN experiment. When train a model, we choose three domains from P, A, C, and S and evaluate on the remaining domain. From Fig.12, we can observe that the POT method was especially better when the model is trained on PAC and evaluated on S, and trained on PAS and evaluated on C. This can be attributed to the feature distribution of sketch and cartoon image in Fig.13. Features of sketch and cartoon images are not much learnable from images from other domains, and it requires tuning the feature extractor. However, other methods show bias here, being unable to perform well.

---

> > > > ### Comment · Reviewer_8wd7 · 2023-11-21
> > > > **Thanks!**
> > > >
> > > > Thanks for the comment. The responses are satisfactory. The thanks for the additional experiments too, I think these will make the paper stronger.
> > > > I am not too convinced on why you don't see expected trends with the ImageNet-C experiments though. Perhaps something that could be investigated further at a later point.
> > > > I have updated my score (5->6, confidence of 2)

---

> > > > > ### Author Response · Authors · 2023-11-22
> > > > > **Thank you for the feedback!**
> > > > >
> > > > > Dear Reviewer 8wd7
> > > > >
> > > > > Thank you for your feedback and raising the score. Your valuable feedback and constructive comments have been immensely helpful for improving our work.
> > > > >
> > > > > As for the ImageNet-C experimental result, we expected the results to some extent. Models pre-trained with the ImageNet dataset can learn general-purpose features. The study that shows last layer retraining of a model trained on ImageNet results in good performance on spurious correlation benchmarks implies the good functionality as feature extractor[3]. In the context of transfer learning, it is shown that the convergence is faster when using ImageNet pretrained models than models randomly initialized, while it does not necessarily improve final target task accuracy.[1][2]. These three findings suggest that parameter saliency methods can identify where in the parameters should be fixed to correct the misclassification, but they are relatively less important because the model already learned good features. This could be the cause of the competitive results appearing in the ImageNet-C comparisons.
> > > > >
> > > > > Reference:
> > > > >
> > > > > [1] Kornblith, Simon, Jonathon Shlens, and Quoc V. Le. "Do better imagenet models transfer better?." Proceedings of the IEEE/CVF conference on computer vision and pattern recognition. 2019.
> > > > > [2]He, Kaiming, Ross Girshick, and Piotr Dollár. "Rethinking imagenet pre-training." Proceedings of the IEEE/CVF International Conference on Computer Vision. 2019.
> > > > > [3] Kirichenko, Polina, Pavel Izmailov, and Andrew Gordon Wilson. "Last layer re-training is sufficient for robustness to spurious correlations." arXiv preprint arXiv:2204.02937 (2022).

---

### Meta-Review · Area_Chair_A9K4 · 2023-12-11

**Metareview:**

This paper builds upon previous research by introducing a novel approach to ranking salient filters, leveraging connections with statistical anomaly detection and extreme value theory. The evaluation spans multiple architectures and datasets, enhancing the generalizability of the proposed method.

However, a significant concern arises from the current form of the paper, where the theoretical proof is deemed confusing. Additionally, the experimental results are criticized for being partial and lacking comparisons against different baselines. Furthermore, the absence of standard deviations in the reported results is a notable limitation, as it hinders the assessment of statistical significance.
While the paper introduces a major contribution by replacing the Gaussian distribution with a heavy-tailed Pareto distribution, the lack of justification for this change both theoretically and empirically is a substantial drawback. The absence of statistically significant experiments raises concerns about the validity and impact of this modification. Furthermore, it is observed that experiments on the CIFAR-10 dataset, as in Levin et al. (2022), are conspicuously absent, casting doubt on the comprehensiveness of the empirical evaluation. Overall, the paper's drawbacks, ranging from clarity issues to experimental limitations, go against accepting the manuscript.

**Justification For Why Not Higher Score:**

The reported results lack rigorous justification for the proposed approach. The absence of standard deviations in the reported results is a notable limitation, as it hinders the assessment of statistical significance.

**Justification For Why Not Lower Score:**

N/A

---

### Decision · Program_Chairs · 2024-01-16

Reject